# How to evaluate your medical time series classification?

## Abstract

Medical time series (MedTS) play a critical role in many healthcare applications, such as vital sign monitoring and the diagnosis of brain and heart diseases. However, the existence of subject-specific features poses unique challenges in MedTS evaluation. Inappropriate evaluation setups that either exploit or overlook these features can lead to artificially inflated classification performance (by up to 50% in accuracy on ADFTD dataset): this concern has received little attention in current research. Here, we categorize the existing evaluation setups into two primary categories: subject-dependent and subject-independent. We show the subject-independent setup is more appropriate for different datasets and tasks. Our theoretical analysis explores the feature components of MedTS, examining how different evaluation setups influence the features that a model learns. Through experiments on six datasets (spanning EEG, ECG, and fNIRS modalities) using four different methods, we demonstrate step-by-step how subject-dependent utilizes subject-specific features as a shortcut for classification and leads to a deceptive high performance, suggesting that the subject-independent setup is more precise and practicable evaluation setup in real-world. This comprehensive analysis aims to establish clearer guidelines for evaluating MedTS models in different healthcare applications. Code to reproduce this work in https://anonymous.4open.science/r/MedTS_Evaluation-733F.

## 1 Introduction

Medical time series (MedTS) are specialized time series data representing continuous recordings of physiological signals from human subjects Wang et al. (2024b), including EEG, ECG, fNIRS, and PPG signals Badr et al. (2024); Liu et al. (2021); Eastmond et al. (2022); Esgalhado et al. (2021).

We propose a taxonomy for MedTS evaluation based on dataset composition and subject dependency. Compared to general time series,

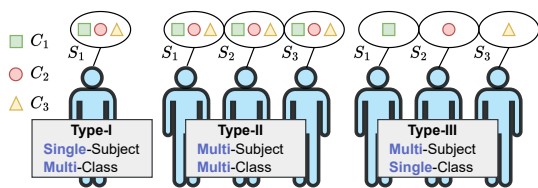

Figure 1: **Types of MedTS Datasets.** S and C denote subject and class, respectively.

MedTS contain additional subject-specific information, as they are physiological data collected from human subjects and are associated with subject IDs alongside the ground truth labels Wang et al. (2024c). This subject-specific information, along with each subject's medical state, allows us to categorize the MedTS dataset into three types based on the number of subjects and the medical classes per subject (Figure 1): single-subject and multi-class, multi-subject and multi-class, and multi-subject and single-class. For simplicity, we denote them as **Type-I**, **Type-II**, and **Type-III** MedTS datasets respectively. Here, the distinction between single/multi-class for a subject indicates whether the subject's medical state is fixed or dynamic over time. See detailed explanations and real-world examples in Section 2.1. Depending on the type of MedTS, various tasks can be designed for real-world medical applications, including health monitoring Jafari et al. (2023), disease diagnosis Kiyasseh et al. (2021), and brain-computer-interface (BCI) Altaheri et al. (2023).

We categorize existing MedTS evaluation setups into two primary groups (Section 2.2, Figure 2): Subject-Dependent Ieracitano et al. (2019); Arif et al. (2024) and Subject-Independent Nath et al. (2020); Kiyasseh et al. (2021). In the **Subject-Dependent** setup, the training, validation, and test

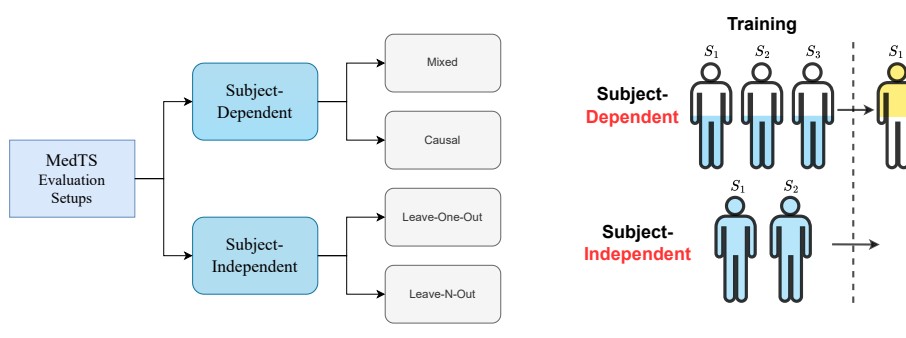

(a) Types of Evaluation Setups      (b) Illustration of Setups Wang et al. (2024b)

Figure 2: **Types of MedTS Evaluation Setups.** (a) This diagram shows the two main evaluation setups and their sub-types, (b) This figure adopted from Wang et al. (2024b) shows the differences between the two main setups: subject-dependent and subject-independent.

sets are split similarly to general time series classification, where samples could be assigned any of the into these three sets. This approach allows samples with the same subject ID to appear in multiple sets simultaneously. In contrast, the **Subject-Independent** setup addresses subject-specific information by splitting the data based on subjects, ensuring that samples from the same subject appear exclusively in either the training, validation, or test set.

However, the inherent subject-specific information in MedTS introduces unique challenges that do not exist in general time series classification problems. As observed in various MedTS classification tasks, such as disease diagnosis, different evaluation setups can lead to widely varying performance outcomes Nath et al. (2020); Seal et al. (2021). For example, the subject-independent setting achieves a 50.65% F1 score on the ADFTD dataset, which is 46.85% lower than the subject-dependent setting Wang et al. (2024c)! A general explanation for these performance differences is that the model overfits subject-specific features instead of learning common features across subjects Arif et al. (2024); Wang et al. (2024b). However, no study has thoroughly quantified how much models depend on subject-specific features in the subject-dependent setup.

Intuitively, researchers should use the subject-independent setup when evaluating MedTS datasets, as it better mirrors real-world conditions where models must generalize to unseen patients. In practice, it is unrealistic and problematic for a model to be trained on data from the same subjects that appear in the test set, as this leads to overfitting on subject-specific features rather than learning patterns that generalize across different subjects. Consequently, the subject-independent setup provides a more rigorous and reliable measure of a model's performance in medical time series classification. However, many studies are still evaluating MedTS in subject-dependent setup, which is problematic.

In this paper, we empirically demonstrate the impact of the subject-dependent setup on learning subject-specific features and how these features influence model performance in Type-III MedTS disease diagnosis tasks. By randomly shuffling the ground truth labels while retaining the original subject IDs, we train models under both the subject-dependent and subject-independent setups. This procedure effectively removes disease-related features, ensuring that the model can only rely on subject-specific features to classify labels. By observing the performance changes, we assess the influence of subject-specific features on overall model performance.

We conduct experiments on six MedTS datasets, including three EEG datasets, two ECG datasets, and one fNIRS dataset. We train four different models on these datasets to broadly verify our assumptions. The results reveal that, for certain datasets such as ADFTD and PTB, the model's performance under the subject-dependent setup barely declined—from 97.56% to 97.12% and 99.90% to 99.69% in F1 score, respectively, using Medformer Wang et al. (2024c)—even after we completely removed the disease-related features across subjects. These results strongly indicate that it is possible for a model to learn virtually nothing about disease-related features while achieving deceptively high performance by leveraging subject-specific features as a shortcut in the subject-dependent setup. Even increasing the number of subjects could not entirely eliminate this effect. For instance, in the larger PTB-XL dataset, the performance drops from 88.02% F1 score to 71.15% when disease-related features are removed under the subject-dependent setup using Medformer. Yet, this score remained much higher

than the 59.97% F1 score obtained in the subject-independent setup. This finding underscores that **subject-specific features can significantly inflate performance metrics, even in datasets with a large number of subjects, emphasizing the importance of using subject-independent setups** for a more realistic evaluation setup in any tasks that aims to learn cross-subject features, such as disease-related features in Type-III MedTS.

## 2 TAXONOMY OF MEDTS DATASETS AND EVALUATION SETUPS

### 2.1 TYPES OF MEDICAL TIME SERIES DATASET

We categorize MedTS datasets into three types below based on the number of subjects and classes:

**(1) Single-Subject and Multi-Class (Type-I).** This type of MedTS is typically used to develop models tailored to a specific subject. Examples include designing brain-computer interfaces for individuals with disabilities Musk et al. (2019); Scherer et al. (2015) or building personalized health monitoring systems (Levett et al., 2023) using wearable devices. In these scenarios, the model focuses on learning from the time-varying classes generated by a single subject's data over time.

**(2) Multi-Subject and Multi-Class (Type-II).** The second type is used to design general models that can adapt to multiple subjects with varying classes. For instance, mental state recognition systems (Jafari et al., 2023; Li et al., 2022; Dai et al., 2019; Zhang et al., 2018) or sleep state classification systems (Eldele et al., 2021; Mousavi et al., 2019) fall under this category. In this case, the MedTS samples from different subjects are associated with dynamically changing classes over time, meaning no fixed class is assigned to each subject.

**(3) Multi-Subject and Single-Class (Type-III).** In the third type, each subject is assigned a single class that remains consistent over time. For example, once a subject is diagnosed with Alzheimer's Disease during the data collection stage, this subject is always classified as an AD patient. This type of MedTS is often used as a low-cost diagnostic method to supplement or replace more traditional medical approaches Miltiadous et al. (2023b); van Dijk et al. (2022); Wagner et al. (2020). It is important to clarify that the term "single-class" here refers to cases where the medical or physiological state of a subject remains fixed over time (or within a short period without significant change). For example, a patient diagnosed with Alzheimer's Disease (AD) will typically remain in that state for decades after the diagnosis. However, a subject may also have multiple coexisting medical conditions, such as both Alzheimer's and Parkinson's Disease. This scenario represents a multi-label problem in the medical domain (Zhou et al., 2023), which falls outside the scope of this work.

### 2.2 MEDTS EVALUATION SETUPS

We categorize the commonly used evaluation setups in existing MedTS classification studies into two main types below (Figure 2). (Seal et al., 2021; Wang et al., 2024b).

**(1) Subject-Dependent.** Samples from different subjects are randomly shuffled and split into training, validation, and test sets, allowing the same subject's data to appear in all three sets. Depending on whether a causal split is applied, the subject-dependent setup can be further divided into two subtypes: **Mixed (random split):** The splitting of samples from each subject does not account for causal relationships; instead, samples are randomly mixed Dai et al. (2019); Sarkar et al. (2022); Seal et al. (2021). This means that stratification occurs without regard for temporal order. **Causal (temporal split):** The split considers the temporal relationship in the time series for each subject, the past samples included in the training set, and future samples placed in the validation and test sets. This causal setup is necessary for time series forecasting tasks Wu et al. (2021); Zhou et al. (2021); Liu et al. (2024) but is rarely adopted in existing time series classification tasks. We discuss the implications of the causal-subject-dependent setup in more detail in Section 4.3.

**(2) Subject-Independent.** The training, validation, and test sets are split by subjects, ensuring that samples from the same subject ID are exclusively included only in one of the three sets. Subject-independent setup can be further divided into two subtypes: **Leave-One-Out:** Leave only one subject for testing, while all other subjects are used for training Zhang et al. (2023); Li et al. (2022); Miltiadous et al. (2023a). Due to the limited sample size of a single subject, cross-validation is generally necessary to avoid biased evaluations. However, this setup does not include a dedicated

Table 1: **Feature Components Utilization of Setups.** The table provides an overview of the feature components utilized by a trained model $f_\theta$ when classifying a given Type-III MedTS sample $x$ during the validation/test stages across five experimental setups, along with their corresponding classification tasks. The setups include subject-dependent(Sub-Dep), subject-independent(Sub-Inep), subject-discrimination(Sub-Disc), random-label subject-dependent(R-Sub-Dep), and random-label subject-independent(R-Sub-Indep). There are three feature components considered: disease-related features $x^d$, subject-specific features $x^s$, and all other features $x^o$. The classification tasks involve predicting either the label $y$ or the subject ID $z$ of a sample.

| | Feature Components Utilization | | | Classification Tasks | |
|---|---|---|---|---|---|
| Setups | $x^d$ (Disease) | $x^s$ (Subject) | $x^o$ (Other) | $y$ (Label) | $z$ (Subject ID) |
| Sub-Dep | ✓ | ✓ | ✓ | ◯ | |
| Sub-Indep | ✓ | | ✓ | ◯ | |
| Sub-Disc | | ✓ | ✓ | | ◯ |
| R-Sub-Dep | | ✓ | ✓ | ◯ | |
| R-Sub-Indep | | | ✓ | ◯ | |

validation set, which can lead to overfitting. **Leave-N-Out:** Leave $N$ subjects for validation and/or test sets Ahmed et al. (2020); Pandey & Seeja (2022); Wang et al. (2024c). In this setup, a larger pool of subjects is used for testing, making it more robust than the leave-one-out approach.

In the evaluation setups, the inclusion of a *validation set* and *cross-validation* are optional but strongly recommended for real-world applications to ensure a comprehensive and reliable evaluation. Here, we do not consider the impact of data imbalance.

## 3 METHOD

*This paper mainly discusses the Type-III MedTS dataset*, characterized by multiple subjects with a single class assigned per subject, as categorized in section 2.1 and illustrated in Figure 1. *For the Type-II MedTS datasets and their evaluation setups, we briefly discuss them in the section 4.3* in experiments later and leave a more detailed analysis to future works. *We discussed Type-I datasets in Appendix C* but did not conduct experiments due to the lack of public datasets with sufficient data from a single subject, which is necessary for Type-I experiments where the training set includes only one subject and requires a large sample size.

In this section, we first introduce our notations and assumptions for describing the feature components of Type-III MedTS. Next, we analyze the subject-dependent and subject-independent setups in the context of feature components utilized in the Type-III MedTS classification tasks. Finally, we propose several new experimental setups based on these two setups to validate our assumptions systematically.

### 3.1 NOTATIONS AND ASSUMPTIONS OF TYPE-III MEDTS DATASET

Given a Type-III MedTS sample $x \in \mathbb{R}^{T \times C}$, where $T$ denotes the number of timestamps and $C$ represents the number of channels, the sample is associated with a corresponding disease-related label $y \in \mathbb{R}^K$. Here, $K$ indicates the number of medically relevant classes, such as different disease types. Each sample is also assigned to a subject ID $z \in \mathbb{R}$, identifying the subject to which it belongs.

In time series forecasting tasks, researchers commonly decompose data into trend and seasonal feature components and design mechanisms to effectively leverage these components for improved representation learning Wu et al. (2021); Wang et al. (2024a); Fraikin et al. (2023). Similarly, Type-III MedTS data also have several distinct feature components. We assume that a Type-III MedTS sample $x$ consists of three feature components:

$$x = x^d + x^s + x^o \tag{1}$$

where $x^d$ represents **disease-related features**, $x^s$ denotes **subject-specific features**, and $x^o$ includes **all other features**, such as noise, artifacts, and all task-irrelevant features. In Type-III MedTS learning for disease diagnosis tasks, our goal is to train a model $f_\theta : x \to y$ that accurately maps a given sample $x$ to its corresponding disease-related label $y$.

However, in the context of disease diagnosis using Type-III MedTS, the choice between subject-dependent and subject-independent setups can result in significant disparities in model perfor-

mance Nath et al. (2020); Seal et al. (2021); Wang et al. (2024b). We aim to learn a model using disease-related features $x^d$ to achieve accurate classification. Ideally, the model $f_\theta$ should rely solely on $x^d$ and be entirely independent of subject-specific features $x^s$ when predicting the label $y$.

Given the unique nature of Type-III MedTS data, where **samples with the same subject ID $z$ always share the same label $y$, but samples with the same label $y$ do not necessarily share the same subject ID $z$**, there is a risk that the model $f_\theta$ might exploit subject-specific features $x^s$ as a shortcut for predicting $y$, rather than learning the disease-related features $x^d$. Although some existing studies have attempted to address this issue and propose methods to tackle it, effectively decomposing $x^d$ and $x^s$ from $x$ remains an open challenge Zhang et al. (2020); Yang et al. (2022); Wang et al. (2024c).

To summarize, we assume that **subject-specific features $x^s$ exist in each sample $x$ within the Type-III MedTS dataset, and it is possible for the trained model $f_\theta$ to utilize $x^s$ as a shortcut for classifying the label $y$ under the subject-dependent setup. This could result in deceptively high performance while the model learns little to nothing about the disease-related features $x^d$.**

### 3.2 Feature Components Utilized in Subject-Dependent vs Independent

We analyze the differences of two setups from the perspective of feature components used when classifying a Type-III MedTS sample $x$. Their feature components utilization are presented in table 1.

**a) Subject-Dependent.** The subject-dependent setup disregards the subject ID, treating the Type-III MedTS classification problem similarly to general time series classification. Samples from different subjects within the dataset are shuffled, mixed, and randomly divided into training, validation, and test sets. Samples sharing the same subject ID can be present inclusively in these three sets.

In this scenario, for a given sample $x$ in the validation/test set, the trained model $f_\theta$ could leverage both its disease-related features $x^d$ and subject-specific features $x^s$ to predict the label $y$. This happens because the model $f_\theta$ gains access to disease-related features by learning from training samples with the same label $y$ and simultaneously acquires subject-specific features by learning from training samples with the same subject ID $z$. Consequently, this setup can lead to an overestimation of the model's performance, as it can exploit subject-specific information rather than generalizing based on disease-related features alone.

**b) Subject-Independent.** The subject-independent setup considers the effect of subject-specific features and closely simulates real-world applications. In this setup, the training, validation, and test sets are split based on subjects rather than individual samples, ensuring that all samples from a particular subject (i.e., with the same subject ID) are exclusively included in one of the three sets.

In this scenario, for any given sample $x$ in the validation/test set, the trained model $f_\theta$ can only leverage the disease-related features $x^d$ to predict the label $y$, as no samples with the same subject ID $z$ are present in the training set. This setup aligns with real-world applications of disease diagnosis using Type-III MedTS, where a label is assigned to an entire subject, and it's impractical to have access to labeled samples from that subject in advance during training.

However, this setup often faces several challenges and can sometimes exhibit poor performance. Firstly, while the subject independent setup prevents model $f_\theta$ from exploiting subject-specific features $x^s$ as a shortcut for classification during validation and testing, it remains possible for the model to rely on subject-specific features during the training stage, impeding its ability to learn task-related subject-invariant features. Secondly, from a domain generalization perspective, different subjects can be considered distinct domains, making it challenging to efficiently extract disease-related features that generalize well across unseen domains (i.e., subjects in the validation/test sets). Currently, there are no effective methods to decompose subject-specific and disease-related features during the training stage fully. This remains an open challenge in the field.

### 3.3 New Experimental Setups to Validate Assumptions

To demonstrate that subject-specific features are universally present in Type-III MedTS datasets, we first introduce the subject-discrimination setup to identify these features. Following this, we design two additional setups—random-label subject-dependent and random-label subject-independent—to manually remove the disease-related features from the dataset. These setups allow us to validate our assumption that models trained under the subject-dependent setup heavily rely on subject-specific

Table 2: The information for the processed datasets. The table shows the number of subjects, samples, classes, channels, sampling rate, sample timestamps, type of MedTS, and file size. Here, **#-Timestamps** indicates the number of timestamps per sample.

| Datasets | #-Subject | #-Sample | #-Class | #-Channel | #-Timestamps | Sampling Rate | MedTS-Type | File Size |
|---|---|---|---|---|---|---|---|---|
| ADFTD | 88 | 69,752 | 3 | 19 | 256 | 256Hz | Type-III | 2.52GB |
| TDBrain | 72 | 6,240 | 2 | 33 | 256 | 256Hz | Type-III | 571MB |
| PTB | 198 | 64,356 | 2 | 15 | 300 | 250Hz | Type-III | 2.15GB |
| PTB-XL | 17,596 | 191,400 | 5 | 12 | 250 | 250Hz | Type-III | 4.28GB |
| EEGMMIDB | 106 | 9236 | 3 | 64 | 640 | 160Hz | Type-II | 2.82GB |
| WG-fNIRS | 26 | 9360 | 2 | 72 | 25 | 10Hz | Type-II | 129MB |

features as a shortcut for label classification. The feature components utilized and classification tasks for three new setups are present in table 1.

**a) Subject-Discrimination.** The disease label $y$ of a subject in a MedTS dataset is assigned by medical experts, ensuring that disease-related features $x^d$ are present in the MedTS samples for that subject. However, there is no guarantee that subject-specific features $x^s$ exist, although researchers usually assume they exist. To investigate the presence of subject-specific features $x^s$ for samples in each subject, we design this setup to prove the existence of subject-specific features by trying to classify the subject ID $z$ of samples.

In this experiment, we randomly shuffle, mix, and split all samples into training, validation, and test sets. We then train a model $f_\theta$ to predict the subject ID $z$ of a given sample $x$. Since the model is not provided with the disease label $y$ during training, and given that the number of subjects is typically much larger than the number of disease labels, the model cannot rely on disease-related features $x^d$ as a shortcut for subject ID classification. **Therefore, if the model performs well in classifying the subject ID $z$, it strongly indicates the existence of subject-specific features $x^s$ within the dataset.**

**b) Random-Label Subject-Dependent.** Previous research has consistently shown that models evaluated under the subject-dependent setup generally achieve much higher performance compared to the subject-independent setup Nath et al. (2020); Arif et al. (2024); Wang et al. (2024c). While it is widely assumed that the model under the subject-dependent setup exploits subject-specific features $x^s$ as a shortcut for classifying the label $y$, the extent of the model's reliance on $x^s$ and whether it still learns disease-related features $x^d$ remains unclear. To investigate the clear influence of subject-specific features $x^s$ on model performance, we introduce the random-label subject-dependent setup.

This setup differs slightly from the original subject-dependent setup. We retain the subject ID but randomly shuffle the label of each subject before splitting the samples into training, validation, and test sets. This operation disrupts the relationship between disease-related features $x^d$ and the label $y$, effectively preventing the model $f_\theta$ from leveraging $x^d$ for prediction.

As a result, in the random-label subject-dependent setup, the trained model $f_\theta$ can rely solely on subject-specific features $x^s$ to predict the label $y$ for any given sample $x$. **By comparing the performance of the random-label subject-dependent setup with the original subject-dependent setup, we can directly assess the extent to which the model relies on subject-specific features $x^s$ as a shortcut and evaluate how much it learns disease-related features $x^d$.**

**c) Random-Label Subject-Independent.** Similar to the random-label subject-dependent setup, we apply the same process in the subject-independent setup, where we randomly shuffle the label of each subject while retaining their subject IDs before splitting the data into training, validation, and test sets. This ensures that the model $f_\theta$ cannot learn any disease-related features $x^d$.

The random-label subject-independent setup serves as a control for the random-label subject-dependent setup, enabling us to isolate the potential impact of other features $x^o$. In this setup, the model is unable to utilize either disease-related features $x^d$ or subject-specific features $x^s$ to classify the label $y$; instead, it can only rely on other features $x^o$. **In other words, if the other features $x^o$ are independent of the label $y$, the performance under the random-label subject-independent setup should be completely random.**

## 4 Experiments

To make the experimental scenarios more robust and pervasive, we train on four different models: Multi-Layer Perception (MLP) Popescu et al. (2009), Temporal Convolutional Neural Networks

(TCN) Bai et al. (2018), vanilla Transformer Vaswani et al. (2017), and Medformer Wang et al. (2024c), to cover different architectures from MLP, CNN, to transformers. The implementation details of four methods are presented in Appendix B. We evaluate six MedTS datasets, including three EEG datasets, ADTFD (Miltiadous et al., 2023b), TDBrain van Dijk et al. (2022), and EEGMMIDB Schalk et al. (2004), and two ECG datasets, PTB PhysioBank (2000) and PTB-XL Wagner et al. (2020), and one fNIRS datasets, WG-fNIRS Shin et al. (2018). The data processing processes are presented in Appendix A. The information of processed datasets is listed in Table 2. We employ two key evaluation metrics: accuracy and F1 score (macro-averaged). The training process is conducted with five random seeds (41-45) to compute the mean and standard deviation of the models. All experiments are run on an NVIDIA RTX 4090 GPU and a server with 4 RTX A5000 GPUs.

Unless otherwise specified in this section, the subject-dependent setup refers to the mix-subject-dependent setup, and the subject-independent setup refers to the leave-N-out subject-independent setup, as categorized in section 2.2. Experimental section 4.1 and 4.2 systematically prove our assumptions of Type-III MedTS datasets analyzed in section 3. We also briefly present some results and findings from evaluating Type-II MedTS datasets using different subject-dependent sub-types to show the influence of the causal-split in section 4.3, leaving more detailed analysis for future work.

## 4.1 RESULTS OF SUBJECT-DEPENDENT AND SUBJECT-INDEPENDENT ON TYPE-III MEDTS

Table 3: Results comparison of Type-III MedTS datasets evaluated under subject-dependent(Sub-Dep) and subject-independent(Sub-Indep) setups. The model's performance under the subject-dependent setup consistently exceeds the subject-independent setup across all datasets and models.

| Models | | MLP | | TCN | | Transformer | | Medformer | |
|---|---|---|---|---|---|---|---|---|---|
| Datasets | Setups | Accuracy | F1 Score | Accuracy | F1 Score | Accuracy | F1 Score | Accuracy | F1 Score |
| ADFTD | Sub-Dep | $58.66_{\pm0.61}$ | $55.48_{\pm0.44}$ | $81.42_{\pm0.49}$ | $80.25_{\pm0.54}$ | $97.00_{\pm0.43}$ | $96.86_{\pm0.44}$ | $\mathbf{97.66_{\pm0.76}}$ | $\mathbf{97.56_{\pm0.79}}$ |
| (3-Classes) | Sub-Indep | $49.10_{\pm1.01}$ | $43.78_{\pm0.28}$ | $50.46_{\pm1.35}$ | $47.32_{\pm1.27}$ | $50.47_{\pm2.14}$ | $48.09_{\pm1.59}$ | $\mathbf{52.37_{\pm1.51}}$ | $\mathbf{48.72_{\pm1.18}}$ |
| TDBrain | Sub-Dep | $81.93_{\pm0.26}$ | $80.16_{\pm0.29}$ | $97.32_{\pm0.28}$ | $97.16_{\pm0.30}$ | $\mathbf{97.60_{\pm0.19}}$ | $\mathbf{97.45_{\pm0.21}}$ | $96.70_{\pm0.42}$ | $96.51_{\pm0.45}$ |
| (2-Classes) | Sub-Indep | $69.42_{\pm0.64}$ | $69.37_{\pm0.64}$ | $83.98_{\pm2.31}$ | $83.93_{\pm2.35}$ | $\mathbf{86.58_{\pm0.76}}$ | $\mathbf{86.52_{\pm0.79}}$ | $83.92_{\pm1.01}$ | $83.69_{\pm1.09}$ |
| PTB | Sub-Dep | $99.80_{\pm0.02}$ | $99.63_{\pm0.03}$ | $\mathbf{99.95_{\pm0.01}}$ | $\mathbf{99.91_{\pm0.03}}$ | $99.92_{\pm0.02}$ | $99.86_{\pm0.04}$ | $99.94_{\pm0.02}$ | $99.90_{\pm0.05}$ |
| (2-Classes) | Sub-Indep | $77.76_{\pm0.46}$ | $70.02_{\pm0.60}$ | $\mathbf{83.97_{\pm2.26}}$ | $\mathbf{78.99_{\pm3.44}}$ | $77.37_{\pm1.02}$ | $68.47_{\pm2.19}$ | $77.86_{\pm1.64}$ | $69.93_{\pm2.69}$ |
| PTB-XL | Sub-Dep | $66.99_{\pm0.10}$ | $52.98_{\pm0.24}$ | $88.61_{\pm0.42}$ | $85.21_{\pm0.73}$ | $87.86_{\pm0.32}$ | $84.50_{\pm0.41}$ | $\mathbf{90.48_{\pm0.24}}$ | $\mathbf{88.02_{\pm0.33}}$ |
| (5-Classes) | Sub-Indep | $66.16_{\pm0.16}$ | $51.13_{\pm0.20}$ | $\mathbf{73.30_{\pm1.00}}$ | $\mathbf{62.10_{\pm0.29}}$ | $71.13_{\pm0.33}$ | $59.58_{\pm0.55}$ | $71.37_{\pm0.44}$ | $59.97_{\pm0.41}$ |

**Setup.** We begin by comparing the results between the subject-dependent and subject-independent setups. In the subject-dependent setup, the training, validation, and test sets are split based on individual samples, allowing samples with the same subject ID to be present in all three sets simultaneously. In contrast, the subject-independent setup splits these sets based on subjects, ensuring that samples with the same subject ID are exclusively included in only one of the three sets. As discussed in Section 3.2, for a given sample $x$, a trained model $f_\theta$ following the subject-dependent setup can leverage both disease-related features $x^d$ and subject-specific features $x^s$ to classify the label $y$. However, in the subject-independent setup, the trained model $f_\theta$ can rely solely on the disease-related features $x^d$ for classification. By comparing the results between these two setups, we can gain insights into how subject-specific features influence the model's performance.

**Results.** The results are summarized in Table 3, where "Sub-Dep" and "Sub-Indep" refer to the subject-dependent and subject-independent setups, respectively. For each dataset, we highlight in bold the best-performing result across the four methods within each setup. Across all datasets, it's evident that the subject-dependent setup consistently outperforms the subject-independent setup, regardless of the training method used. The performance gap between the subject-dependent and subject-independent setups varies across different datasets. For instance, when using the Medformer model, the F1 score difference between two setups is approximately 50% F1 score for the ADFTD dataset, but only around 13% F1 score for the TDBrain dataset. This variation suggests that the ratio of disease-related features $x^d$ to subject-specific features $x^s$ differs across datasets.

## 4.2 RESULTS OF THREE NEW EXPERIMENTAL SETUPS ON TYPE-III MEDTS

In this section, we perform experiments using three newly designed setups: subject-discrimination, random-label subject-dependent, and random-label subject-independent. These experiments aim to analyze and verify the assumptions outlined in the method section 3.

### 4.2.1 RESULTS OF SUBJECT-DISCRIMINATION

Table 4: The results of Type-III datasets evaluated under the subject-discrimination(Sub-Disc) setup are summarized in the table. This setup aims to verify the existence of subject-specific features in the datasets. The high performance observed across all four datasets supports this assumption, indicating that subject-specific features $x^s$ are present.

| Models | | MLP | | TCN | | Transformer | | Medformer | |
|---|---|---|---|---|---|---|---|---|---|
| Datasets | Setups | Accuracy | F1 Score | Accuracy | F1 Score | Accuracy | F1 Score | Accuracy | F1 Score |
| ADFTD (88-Subjects) | Sub-Disc | 30.89±0.66 | 30.90±0.61 | 86.31±1.60 | 86.10±1.59 | 98.15±0.49 | 98.00±0.52 | **98.78±0.16** | **98.72±0.17** |
| TDBrain (72-Subjects) | Sub-Disc | 29.78±0.73 | 28.83±0.73 | 58.23±4.03 | 56.92±4.42 | 80.93±1.78 | 80.14±1.83 | **86.93±1.50** | **86.56±1.58** |
| PTB (198-Subjects) | Sub-Disc | 99.16±0.02 | 99.21±0.05 | **99.68±0.03** | **99.68±0.03** | 99.54±0.03 | 99.60±0.04 | 99.64±0.04 | 99.66±0.04 |
| PTB-XL (17,596-Subjects) | Sub-Disc | 5.32±0.13 | 4.68±0.11 | **84.04±2.39** | **84.35±2.18** | 69.92±1.51 | 68.01±1.67 | 78.27±0.50 | 76.64±0.53 |

**Setup.** The subject-discrimination setup is similar to the subject-dependent setup in that we randomly shuffle, mix, and split all samples into training, validation, and test sets, allowing samples with the same subject ID to be present in all three sets simultaneously. However, the classification task in this setup is shifted from predicting the label $y$ to identifying the subject ID $z$. The primary goal of this setup is to confirm the existence of subject-specific features $x^s$ within a dataset and to assess their strength based on the model's classification performance. As discussed in Section 3.3, when validating or testing a given sample $x$, the trained model $f_\theta$ is unable to rely on disease-related features $x^d$ as shortcuts for subject ID $z$ classification, since there is typically a one-to-many relationship between disease label $y$ and subject ID $z$. In rare cases where a dataset of two subjects has opposite labels, the model $f_\theta$ might theoretically use disease-related features $x^d$ as a shortcut to classify the subject ID $z$. However, this scenario does not apply to the datasets used in this paper, where the number of subjects far exceeds the number of labels.

**Results.** The results of the subject-discrimination setup are presented in Table 4. The highest F1 scores across the four methods are 98.72%, 86.56%, 99.68%, and 84.35% for the ADFTD, TDBrain, PTB, and PTB-XL datasets, respectively. The performance for ADFTD and PTB is remarkably high, approaching an F1 score of 100%, indicating that these two datasets has very strong subject-specific features. Even for the larger PTB-XL dataset, which includes 17,596 subjects, the model still achieved an F1 score exceeding 80%. Overall, these results provide compelling evidence for the existence of subject-specific features across all four datasets, with some datasets, like ADFTD and PTB, exhibiting very strong subject-specific features, while others, like TDBrain and PTB-XL, display relatively weaker but still significant subject-specific characteristics.

### 4.2.2 RESULTS OF RANDOM-LABEL SUBJECT-DEPENDENT

Table 5: The results of Type-III MedTS datasets evaluated under the random-label subject-dependent(R-Sub-Dep). For comparison, we also include the results of the subject-dependent(Sub-Dep) and subject-discrimination(Sub-Disc) setups in the table. The negligible performance drop from the subject-dependent to the random-label subject-dependent setup of ADFTD and PTB indicates that the model learns almost nothing about disease-related features $x_d$ during training under the subject-dependent setup for these two datasets.

| Models | | MLP | | TCN | | Transformer | | Medformer | |
|---|---|---|---|---|---|---|---|---|---|
| Datasets | Setups | Accuracy | F1 Score | Accuracy | F1 Score | Accuracy | F1 Score | Accuracy | F1 Score |
| ADFTD (3-Classes) | Sub-Dep | 58.66±0.61 | 55.48±0.44 | 81.42±0.49 | 80.25±0.54 | 97.00±0.43 | 96.86±0.44 | **97.66±0.76** | **97.56±0.79** |
| | R-Sub-Dep | 46.33±1.48 | 45.88±1.41 | 75.16±1.68 | 75.00±1.82 | 96.99±0.29 | 96.97±0.29 | **97.14±0.74** | **97.12±0.74** |
| | Sub-Disc | 30.89±0.66 | 30.90±0.61 | 86.31±1.60 | 86.10±1.59 | 98.15±0.49 | 98.00±0.52 | **98.78±0.16** | **98.72±0.17** |
| TDBrain (2-Classes) | Sub-Dep | 81.93±0.26 | 80.16±0.29 | 97.32±0.28 | 97.16±0.30 | **97.60±0.19** | **97.45±0.21** | 96.70±0.42 | 96.51±0.45 |
| | R-Sub-Dep | 63.40±2.22 | 62.42±2.29 | **89.88±2.26** | **89.71±2.21** | 86.88±1.26 | 86.65±1.35 | 86.17±2.71 | 85.95±2.72 |
| | Sub-Disc | 29.78±0.73 | 28.83±0.73 | 58.23±4.03 | 56.92±4.42 | 80.93±1.78 | 80.14±1.83 | **86.93±1.50** | **86.56±1.58** |
| PTB (2-Classes) | Sub-Dep | 99.80±0.02 | 99.63±0.03 | **99.95±0.01** | **99.91±0.03** | 99.92±0.02 | 99.86±0.04 | 99.94±0.02 | 99.90±0.05 |
| | R-Sub-Dep | 99.44±0.02 | 99.44±0.03 | **99.83±0.03** | **99.83±0.03** | 99.64±0.11 | 99.64±0.11 | 99.69±0.04 | 99.69±0.04 |
| | Sub-Disc | 99.16±0.02 | 99.21±0.05 | **99.68±0.03** | **99.68±0.03** | 99.54±0.03 | 99.60±0.04 | 99.64±0.04 | 99.66±0.04 |
| PTB-XL (5-Classes) | Sub-Dep | 66.99±0.10 | 52.98±0.24 | 88.61±0.42 | 85.21±0.73 | 87.86±0.32 | 84.50±0.41 | **90.48±0.24** | **88.02±0.33** |
| | R-Sub-Dep | 21.62±0.15 | 21.58±0.13 | **72.55±0.39** | **72.54±0.39** | 62.53±2.07 | 62.52±2.07 | 71.16±1.58 | 71.15±1.58 |
| | Sub-Disc | 5.32±0.13 | 4.68±0.11 | **84.04±2.39** | **84.35±2.18** | 69.92±1.51 | 68.01±1.67 | 78.27±0.50 | 76.64±0.53 |

**Setup.** The random-label subject-dependent setup builds upon the subject-dependent setup with an additional step. Before shuffling and splitting all samples into training, validation, and test sets, we first randomly assign a new label to each subject. For instance, in the ADFTD dataset, which has three classes (0, 1, and 2) representing different medical conditions, we randomly assign each subject a new label from these three classes, replacing their true label. The purpose of this setup is to deliberately mask the disease-related features in the dataset, ensuring that the trained model $f_\theta$ theoretically cannot learn any disease-related features during training. As a result, the model can only rely on subject-specific features to classify the label $y$ of a given sample $x$ during the validation and test stages. The performance drop observed in this setup compared to the original subject-dependent setup will indicate the extent to which the model learned disease-related features in the raw subject-dependent setup.

**Results.** The results of the random-label subject-dependent setup are presented in Table 5. For comparison, we also include the results from the original subject-dependent and subject-discrimination setups. The findings are both intriguing and surprising. The best results among the four methods in the random-label subject-dependent setup are 97.12%, 89.71%, 99.83%, and 72.54% for the ADFTD, TDBrain, PTB, and PTB-XL datasets, respectively, which are unexpectedly high.

Considering that there are no disease-related features available under this setup, the model's performance should, in theory, be completely random if it relies solely on disease-related features $x^d$ to classify the label $y$. However, taking the Medformer results as an example, the performance drop from the subject-dependent to the random-label subject-dependent setup is almost negligible for both ADFTD and PTB. This indicates that the model learns almost nothing about disease-related features $x^d$ during training in the subject-dependent setup and instead relies entirely on subject-specific features $x^s$ as a shortcut for label $y$ classification. These findings are consistent with the results from the subject-discrimination setup, where ADFTD and PTB also achieved high classification performance, demonstrating their strong subject-specific features. For the TDBrain and PTB-XL datasets, which show relatively weaker subject-specific features in the subject-discrimination setup, the performance drop from subject-dependent to random-label subject-dependent using Medformer is approximately 10% and 17% F1 score, respectively. This indicates that the model does, in fact, learn some disease-related features $x^d$ in the subject-dependent setup for these two datasets.

In summary, these results suggest that, depending on the dataset, it is possible for the model $f_\theta$ to achieve a very high F1 score (over 95%) under the subject-dependent setup while learning little to nothing about the meaningful disease-related features $x^d$ in the dataset. One might assume that increasing the number of subjects could help the model focus more on general disease-related features $x^d$ rather than relying on subject-specific features $x^s$ as shortcuts. However, even with the large PTB-XL dataset containing 17,596 subjects, subject-specific features still had a significant impact on performance. For instance, Medformer's performance in the random-label subject-dependent setup reached an F1 score of 71.15%, which easily surpassed the 59.97% F1 score achieved in the subject-independent setup. Therefore, we strongly recommend that researchers avoid using the subject-dependent setup for disease diagnosis tasks when working with Type-III MedTS datasets. Instead, adopting the subject-independent setup provides a more realistic evaluation that aligns with real-world applications.

### 4.2.3 RESULTS OF RANDOM-LABEL SUBJECT-INDEPENDENT

For completeness, we perform the random-label subject-independent setup to confirm that features $x^o$, other than disease-related $x^d$ and subject-specific $x^s$ features, do not significantly affect the model's performance. Appendix D.1 presents the detailed setup description and results.

### 4.3 RESULTS OF DIFFERENT SETUPS ON TYPE-II MEDTS

**Setup.** The causal-subject-dependent setup involves splitting each subject's samples in a time-stratified manner, where the data is divided into time-based segments for training, validation, and test sets. This setup evaluates the model's ability to handle temporal shifts within the data. More critically, many MedTS datasets, due to their limited size, use overlapping samples to increase the number of available samples. In the mixed-subject-dependent setup, these overlapping samples can be split between training and test sets, leading to significant data leakage and artificially inflated performance. The causal-segment setup mitigates this issue by ensuring that time-adjacent samples

Table 6: This table presents the results of Type-II MedTS datasets evaluated under the mixed-subject-dependent (M-Sub-Dep), causal-subject-dependent (C-Sub-Dep), and subject-independent (Sub-Indep) setups. The dramatic performance drop observed for the WG-fNIRS dataset when shifting from the mixed to the causal subject-dependent setup highlights how inflated performance can be mitigated using the causal-subject-dependent setup, especially when overlapping samples are used in data preprocessing. Besides, the performance gap between the causal-subject-dependent and subject-independent setups is relatively small, suggesting that the causal-subject-dependent provides a reliable alternative for classifying Type-II MedTS if the subject-independent setup is not feasible.

| Models | | MLP | | TCN | | Transformer | | Medformer | |
|---|---|---|---|---|---|---|---|---|---|
| Datasets | Setups | Accuracy | F1 Score | Accuracy | F1 Score | Accuracy | F1 Score | Accuracy | F1 Score |
| EEGMMIDB (3-Classes) | M-Sub-Dep | 64.12 ±0.93 | 60.74 ±0.96 | 68.18 ±0.65 | 64.04 ±0.71 | **71.14** ±**0.55** | **68.64** ±**0.47** | 69.82 ±0.59 | 67.08 ±0.52 |
| | C-Sub-Dep | 62.97 ±0.11 | 58.63 ±0.30 | 66.06 ±0.37 | 59.70 ±1.00 | 68.98 ±0.67 | 65.56±0.63 | **69.43** ±**0.72** | 65.50 ±0.73 |
| | Sub-Indep | 61.58±1.01 | 57.98 ±0.98 | 62.80 ±0.67 | 57.58 ±0.87 | 67.32 ±0.65 | 64.03 ±0.53 | **67.59** ±**0.44** | **64.23** ±**0.19** |
| WG-fNIRS (2-Classes) | M-Sub-Dep | 61.35 ±0.45 | 61.32 ±0.47 | 79.75 ±0.66 | 79.74 ±0.66 | **83.74** ±**0.77** | **83.73** ±**0.77** | 72.79 ±1.45 | 72.77 ±1.44 |
| | C-Sub-Dep | 57.16 ±1.60 | 57.10 ±1.59 | **59.64** ±**1.07** | **59.40** ±**0.97** | 59.06 ±0.79 | 59.01 ±0.80 | 59.00 ±1.50 | 58.89 ±1.52 |
| | Sub-Indep | 54.21 ±0.71 | 54.17 ±0.75 | 55.00 ±1.52 | 54.92 ±1.57 | **55.28** ±**0.44** | **55.00** ±**0.52** | 54.88 ±0.65 | 54.82 ±0.66 |

remain together, except at the boundary where the data is split, thereby preventing most overlap between training and test sets. This approach reduces the risk of data leakage and ensures that the model's performance reflects its ability to generalize across temporally shifted data, rather than being inflated by overlapping samples.

**Results.** The results of the causal-subject-dependent setup are presented in Table 6. After preprocessing, the EEGMMIDB dataset contains no overlapping samples, while the WG-fNIRS dataset has 80% overlapping samples. For the Transformer model on EEGMMIDB, the performance gap in F1 score between the mixed-subject-dependent and causal-segment subject-dependent setups is relatively small, around 3%. However, for WG-fNIRS, the performance gap is substantial, approximately 25%. This significant discrepancy underscores the risk of data leakage in datasets with overlapping samples, validating our concern. Therefore, when a subject-independent setup is not feasible, we recommend practitioners adopt the causal-subject-dependent setup, especially for datasets with overlapping samples, to avoid artificially inflated performance results.

## 5 DISCUSSION AND CONCLUSION

**Limitations and Future Works.** While this paper conducts thorough experiments and provides valuable insights into evaluating models on Type-III (multi-subject, single-class) MedTS datasets, we are aware of several limitations. (1) Our experiments were primarily focused on Type-III datasets, and we only conducted preliminary experiments on Type-II (multi-subject, multi-class) datasets, and discussed Type-I (single-subject, multi-class) datasets in Appendix C. Given the distinct nature of these dataset types, future work should explore the behavior of models on Type-I and Type-II MedTS datasets to assess whether the same trends observed here, particularly the reliance on subject-specific information, hold across more personalized or multi-class scenarios. (2) While we have shown that subject-dependent setups can yield misleadingly high performance and advised researchers to avoid them, developing a robust model for subject-independent setups remains a significant challenge. Ensuring models can generalize effectively across different subjects without relying on subject-specific information is a complex task that requires further research and innovation.

**Conclusion.** We introduce a new taxonomy for MedTS based on datasets and subject dependency. Additionally, we propose a decomposition paradigm for MedTS data, consisting of disease-related features $x^d$, subject-specific features $x^s$, and extraneous features $x^o$. To investigate the influence of subject-specific features, we design three new experimental setups, namely subject-discrimination, random-label subject-dependent, and random-label subject-independent, to demonstrate their existence and quantify their impact on model performance. Our experiments reveal that it is possible for a model to learn little to nothing about disease-related features $x^d$ while achieving deceptively high performance under the subject-dependent setup. These findings suggest that the subject-independent setup is a more practical setup for MedTS evaluation.

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

## APPENDIX A   DATA PREPROCESSING

### A.1   TDBRAIN PREPROCESSING

The TDBrain dataset[1], referenced in the paper van Dijk et al. (2022), is a large permission-accessible EEG time series dataset recording brain activities of 1274 subjects with 33 channels. Each subject has two trials: one under eye open and one under eye closed setup. The dataset includes a total of 60 labels, with each subject potentially having multiple labels indicating multiple diseases simultaneously. In this paper, we utilize a subset of this dataset containing 25 subjects with Parkinson's disease and 25 healthy controls, all under the eye-closed task condition. Each eye-closed trial is segmented into non-overlapping 1-second samples with 256 timestamps, and any samples shorter than 1-second are discarded. This process results in 6,240 samples. Each sample is assigned a subject ID to indicate its originating subject. For the training, validation, and test set splits, we employ the subject-independent setup. Samples with subject IDs {18,19,20,21,46,47,48,49} are assigned to the validation set, while samples with subject IDs {22,23,24,25,50,51,52,53} are assigned to the test set. The remaining samples are allocated to the training set.

### A.2   ADFTD PREPROCESSING

The **A**lzheimer's **D**isease and **F**rontotemporal **D**ementia (ADFTD) dataset[2], referenced in the papers Miltiadous et al. (2023b;a), is a public EEG time series dataset with 3 classes, including 36 Alzheimer's disease (AD) patients, 23 Frontotemporal Dementia (FD) patients, and 29 healthy control (HC) subjects. The dataset has 19 channels, and the raw sampling rate is 500Hz. Each subject has a trial, with trial durations of approximately 13.5 minutes for AD subjects (min=5.1, max=21.3), 12 minutes for FTD subjects (min=7.9, max=16.9), and 13.8 minutes for HC subjects (min=12.5, max=16.5). A bandpass filter between 0.5-45Hz is applied to each trial. We downsample each trial to 256Hz and segment them into non-overlapping 1-second samples with 256 timestamps, discarding any samples shorter than 1 second. This process results in 69,752 samples. For the training, validation, and test set splits, we employ both the subject-dependent and subject-independent setups. For the subject-dependent setup, we allocate 60%, 20%, and 20% of total samples into the training, validation, and test sets, respectively. For the subject-independent setup, we allocate 60%, 20%, and 20% of total subjects with their corresponding samples into the training, validation, and test sets, respectively.

### A.3   PTB PREPROCESSING

The PTB dataset[3], referenced in the paper PhysioBank (2000), is a public ECG time series recording from 290 subjects, with 15 channels and a total of 8 labels representing 7 heart diseases and 1 health control. The raw sampling rate is 1000Hz. For this paper, we utilize a subset of 198 subjects, including patients with Myocardial infarction and healthy control subjects. We first downsample the sampling frequency to 250Hz and normalize the ECG signals using standard scalers. Subsequently, we process the data into single heartbeats through several steps. We identify the R-Peak intervals across all channels and remove any outliers. Each heartbeat is then sampled from its R-Peak position, and we ensure all samples have the same length by applying zero padding to shorter samples, with the maximum duration across all channels serving as the reference. This process results in 64,356 samples. For the training, validation, and test set splits, we employ the subject-independent setup. Specifically, we allocate 60%, 20%, and 20% of the total subjects, along with their corresponding samples, into the training, validation, and test sets, respectively.

### A.4   PTB-XL PREPROCESSING

The PTB-XL dataset[4], referenced in the paper Wagner et al. (2020), is a large public ECG time series dataset recorded from 18,869 subjects, with 12 channels and 5 labels representing 4 heart diseases and 1 healthy control category. Each subject may have one or more trials. To ensure consistency, we discard subjects with varying diagnosis results across different trials, resulting in 17,596 subjects

---

[1] https://brainclinics.com/resources/
[2] https://openneuro.org/datasets/ds004504/versions/1.0.6
[3] https://physionet.org/content/ptbdb/1.0.0/
[4] https://physionet.org/content/ptb-xl/1.0.3/

remaining. The raw trials consist of 10-second time intervals, with sampling frequencies of 100Hz and 500Hz versions. For our paper, we utilize the 500Hz version, then we downsample to 250Hz and normalize using standard scalers. Subsequently, each trial is segmented into non-overlapping 1-second samples with 250 timestamps, discarding any samples shorter than 1 second. This process results in 191,400 samples. For the training, validation, and test set splits, we employ the subject-independent setup. Specifically, we allocate 60%, 20%, and 20% of the total subjects, along with their corresponding samples, into the training, validation, and test sets, respectively.

### A.5  WG-fNIRS Preprocessing

The Word Generation functional near-infrared spectroscopy (WG-fNIRS) dataset, referenced in the papers Shin et al. (2018; 2016); Blankertz et al. (2010), contains 26 subjects, each participating in 60 trials across three sessions. In each session, 20 trials were conducted in random order, balanced between two classes: 10 Word Generation (WG) and 10 Baseline (BL) trials. The fNIRS data were recorded from 72 channels at a sampling rate of 10 Hz. Each trial consists of a 2-second instruction period, a 10-second task period, and a 13–15-second resting period. Before further preprocessing, the data are downsampled to 5 Hz. We then segment each trial into 5-second samples with 80% overlap, resulting in 9,360 samples. Each sample is labeled with the subject ID to indicate its originating subject. For the training, validation, and test set splits, we employ mixed subject-dependent, causal subject-dependent, and subject-independent setups. In the mixed subject-dependent setup, 60%, 20%, and 20% of the total samples are randomly assigned to the training, validation, and test sets, respectively. In the causal subject-dependent setup, 60%, 20%, and 20% of each subject's samples, sorted by time order, are allocated to the training, validation, and test sets. In the subject-independent setup, 16, 6, and 6 subjects are randomly assigned to the training, validation, and test sets, respectively.

### A.6  EEGMMIDB Preprocessing

The EEG Motor Movement/Imagery Database (EEGMMIDB), referenced in the paper Schalk et al. (2004); Goldberger et al. (2000), contains recordings from 109 subjects engaged in motor imagery tasks, where they were instructed to either perform or imagine specific movements. Each subject participated in 14 distinct tasks, encompassing both motor execution and motor imagery involving hand and foot movements. The EEG data were recorded from 64 channels at a sampling rate of 160 Hz. Each trial comprises a 2-second preparation phase, during which the subject receives instructions about the task, followed by a 4-second period where the movement is either executed or imagined.

During preprocessing, data from subjects 88, 92, and 100 were removed due to a different sampling rate. For our experiments, we selected task 3, task 7, and task 11, and segmented the 4-second task period, resulting in 9,236 samples. For the training, validation, and test set splits, we employ mixed subject-dependent, causal subject-dependent, and subject-independent setups. In the mixed subject-dependent setup, one-third of the total samples are randomly assigned to each of the training, validation, and test sets. In the causal subject-dependent setup, the samples from task 3, task 7, and task 11 of each subject are assigned to the training, validation, and test sets, respectively. For the subject-independent setup, 36, 36, and 35 subjects are randomly allocated to the training, validation, and test sets, respectively.

## Appendix B  Implementation Details

We implement the baselines based on the Time-Series-Library project[5] from Tsinghua University Wu et al. (2022). The four baselines we employ are MLP Popescu et al. (2009), TCN Bai et al. (2018), Transformer Vaswani et al. (2017), and Medformer Wang et al. (2024c).

For MLP, we employ 2 layers with hidden dimension 256. For Transformer and Medformer, we employ 6 layers for the encoder, with the self-attention dimension $D$ set to 128 and the hidden dimension of the feed-forward networks set to 256. The patch list for Medformer is set to {2,4,8} as default. For TCN, we employ 6 layers for the encoder, with hidden dimension 128 and kernel size 3.

---

[5]https://github.com/thuml/Time-Series-Library

## APPENDIX C   DISCUSSION ON TYPE-I MEDTS

In certain MedTS classification tasks, such as personalized health monitoring or brain-computer interface (BCI) systems, where the goal is to train a model that can efficiently control a device (e.g., a wheelchair) for a specific individual, the subject-dependent setup becomes reasonable. In these cases (Type-I MedTS datasets) the training, validation, and test sets are derived from the same subject. This setup aligns with the task's objective of personalizing the model to perform optimally for a single subject, where cross-subject generalization is not the primary focus. We also emphasize the importance of maintaining temporal order in such classification tasks. The evaluation of such Type-I cases is an important future direction.

## APPENDIX D   MORE RESULTS

### D.1   MORE RESULTS

Table 7: The table presents the results of Type-III datasets evaluated under the random-label subject-independent setup (R-Sub-Indep), along with the results of the original subject-independent setup (Sub-Indep) for comparison. As expected, the performance dropped to random under the random-label subject-independent setup, showing that other features $x_o$ do not impact the model's performance.

| Models | | MLP | | TCN | | Transformer | | Medformer | |
|---|---|---|---|---|---|---|---|---|---|
| Datasets | Setups | Accuracy | F1 Score | Accuracy | F1 Score | Accuracy | F1 Score | Accuracy | F1 Score |
| ADFTD (3-Classes) | Sub-Indep | $49.10_{\pm1.01}$ | $43.78_{\pm0.28}$ | $50.46_{\pm1.35}$ | $47.32_{\pm1.27}$ | $50.47_{\pm2.14}$ | $48.09_{\pm1.59}$ | $\mathbf{52.37_{\pm1.51}}$ | $\mathbf{48.72_{\pm1.18}}$ |
| | R-Sub-Indep | $30.84_{\pm1.99}$ | $29.46_{\pm1.39}$ | $\mathbf{33.25_{\pm6.53}}$ | $\mathbf{31.82_{\pm7.08}}$ | $30.92_{\pm5.25}$ | $29.70_{\pm6.21}$ | $30.67_{\pm7.05}$ | $29.82_{\pm7.53}$ |
| TDBrain (2-Classes) | Sub-Indep | $69.42_{\pm0.64}$ | $69.37_{\pm0.64}$ | $83.98_{\pm2.31}$ | $83.93_{\pm2.35}$ | $\mathbf{86.58_{\pm0.76}}$ | $\mathbf{86.52_{\pm0.79}}$ | $83.92_{\pm1.01}$ | $83.69_{\pm1.09}$ |
| | R-Sub-Indep | $52.48_{\pm6.23}$ | $50.64_{\pm5.77}$ | $55.71_{\pm10.01}$ | $52.55_{\pm9.46}$ | $\mathbf{59.44_{\pm6.60}}$ | $\mathbf{57.21_{\pm6.50}}$ | $58.71_{\pm5.45}$ | $56.82_{\pm4.94}$ |
| PTB (2-Classes) | Sub-Indep | $77.76_{\pm0.46}$ | $70.02_{\pm0.60}$ | $\mathbf{83.97_{\pm2.26}}$ | $\mathbf{78.99_{\pm3.44}}$ | $77.37_{\pm1.02}$ | $68.47_{\pm2.19}$ | $69.93_{\pm2.69}$ | $69.93_{\pm2.69}$ |
| | R-Sub-Indep | $50.27_{\pm6.39}$ | $49.91_{\pm6.34}$ | $\mathbf{50.54_{\pm5.28}}$ | $49.93_{\pm5.45}$ | $48.38_{\pm2.18}$ | $48.16_{\pm1.99}$ | $50.38_{\pm2.58}$ | $\mathbf{49.98_{\pm2.49}}$ |
| PTB-XL (5-Classes) | Sub-Indep | $66.16_{\pm0.16}$ | $51.13_{\pm0.20}$ | $\mathbf{73.30_{\pm1.00}}$ | $\mathbf{62.10_{\pm0.29}}$ | $71.13_{\pm0.33}$ | $59.58_{\pm0.55}$ | $71.37_{\pm0.44}$ | $59.97_{\pm0.41}$ |
| | R-Sub-Indep | $20.00_{\pm0.19}$ | $\mathbf{19.94_{\pm0.16}}$ | $19.76_{\pm0.49}$ | $19.65_{\pm0.55}$ | $\mathbf{20.18_{\pm0.45}}$ | $19.89_{\pm0.39}$ | $19.89_{\pm0.21}$ | $19.79_{\pm0.18}$ |

**Setup.** Similar to the random-label subject-dependent setup, we apply the same process in the subject-independent setup, where we randomly assign a new label to each subject while retaining their subject IDs before splitting the data into training, validation, and test sets. The primary goal of this setup is to eliminate the potential influence of other features $x^o$ on model performance. By masking the disease-related features $x^d$ during training and ensuring that the model $f_\theta$ cannot use subject-specific features $x^s$ as shortcuts under the subject-independent setup during validation and testing, we aim to determine whether any other features $x^o$ besides disease-related and subject-specific features have any impact on model performance. In theory, if the results are completely random under this setup, it would indicate that no other features significantly affect the model's performance.

**Results.** The results of the random-label subject-independent setup are presented in Table 7. As expected, the results for all four datasets are nearly random, corresponding to the number of classes in each dataset. For instance, while some methods, such as the Transformer on the TDBrain dataset, achieved a slightly higher F1 score of 57.21% (compared to the random 50% baseline), this is likely due to the relatively limited number of samples in TDBrain compared to the other three datasets. We consider this slight deviation acceptable. Overall, these results confirm that features other than disease-related and subject-specific features do not affect the model's performance, reinforcing the conclusion that the subject-specific and disease-related features are the primary contributors to classification accuracy in MedTS datasets.

