# OpenReview forum: "How To Evaluate Your Medical Time Series Classification?"
_ICLR.cc/2025/Conference — ICLR 2025 Conference Withdrawn Submission_

### Official Review · Reviewer_Kirt · 2024-10-29

**Soundness:** 3
**Presentation:** 3
**Contribution:** 2
**Rating:** 6
**Confidence:** 4

**Summary:**

This paper addresses the evaluation of models for medical time series (MedTS) classification by comparing subject-dependent and subject-independent setups where models can achieve deceptively high performance by relying on subject-specific features rather than disease-related patterns.
The authors demonstrate this issue using Type-III datasets (multi-subject, single-class) and propose new experimental setups to analyze feature reliance.
Through experiments on six datasets using different models, the authors show that subject-independent setups provide a more realistic measure of performance. The paper offers new experimental setups to analyze feature reliance and provides actionable recommendations for improving MedTS evaluation practices.

**Strengths:**

The paper effectively identifies the issue with subject-dependent setups, showing how they can mislead performance evaluation by exploiting subject-specific features.

The introduction of new experimental setups, subject-discrimination and random-label setups adds depth to the analysis and provides valuable tools for future research.

The experiments span multiple datasets (EEG, ECG, fNIRS) and models (MLP, TCN, Transformer, Medformer), supporting the generalizability of the findings across different modalities.

The work challenges common practices and provides practical advice for researchers working with medical datasets relevant to real world scenarios.

**Weaknesses:**

1. The use of accuracy as a key metric is problematic, especially with some datasets being unbalanced. More appropriate metrics for medical applications, such as AUROC or sensitivity, would provide a better evaluation.

2. The comparison between subject-dependent and subject-independent setups is complicated by different data splitting strategies, making it harder to draw fair conclusions.

3. While the paper introduces novel setups, it overlooks several prior works in the medical field that already compare subject-dependent and subject-independent evaluations. Engaging with this literature would provide important context.

4. The experiments mainly focus on Type-III datasets, leaving questions about how these findings apply to Type-I and Type-II datasets.

More ideas on how to improve performance under subject-independent setups would increase the paper’s practical value.
Exploring the other dataset types (Type-I and Type-II) in future work would make the findings more widely applicable.
Addressing the issue of imbalanced datasets with better metrics would improve the rigor of the experiments.

**Questions:**

1. Why did you choose accuracy over more suitable metrics like sensitivity, specificity, or AUROC for medical datasets?

2. How do you plan to ensure a fairer comparison between subject-dependent and subject-independent setups given the different splitting strategies?

3. Did you look into other works that have explored these two setups in the medical field?

4. How do you anticipate your findings will translate to Type-I and Type-II datasets, where tasks and dependencies are different?

---

### Official Review · Reviewer_Lc4i · 2024-10-31

**Soundness:** 3
**Presentation:** 3
**Contribution:** 1
**Rating:** 3
**Confidence:** 4

**Summary:**

The authors, through extensive experimentation, highlight the importance of excluding the same subjects from both the training and testing dataset splits when assessing various methods in Medical Time Series analysis. Medical data possess unique patterns specific to each subject, which the model learns. If the same subjects (albeit different instances) are used for evaluation, the resulting metrics are artificially inflated because the model has already encountered these patients during its training. Consequently, this performance level is not applicable to real-world situations.

**Strengths:**

- The authors clearly explain the problem of not splitting the train/test dataset patient-wise.
- The manuscript is clear and understandable.
- The evaluation is robust and provides evidence for the hypotheses stated.

**Weaknesses:**

In my opinion, the contribution of this study does not add knowledge to the community. Field studies in this community take this contribution for granted when evaluating their methods. Take the following as some examples.

[1], Attia ZI, Noseworthy PA, Lopez-Jimenez F, Asirvatham SJ, Deshmukh AJ, Gersh BJ, Carter RE, Yao X, Rabinstein AA, Erickson BJ, Kapa S, Friedman PA. An artificial intelligence-enabled ECG algorithm for the identification of patients with atrial fibrillation during sinus rhythm: a retrospective analysis of outcome prediction. Lancet. 2019 Sep 7;394(10201):861-867. doi: 10.1016/S0140-6736(19)31721-0. Epub 2019 Aug 1. PMID: 31378392.

[2] Raghu A., Chandak P., Ridwan A., Guttag J., Stultz CM. Sequential Multi-Dimensional Self-Supervised Learning for Clinical Time Series. International Conference on Machine Learning (ICML), PMLR 202:28531-28548, 2023

[3] Thapa, Rahul, et al. "SleepFM: Multi-modal Representation Learning for Sleep Across Brain Activity, ECG and Respiratory Signals." arXiv preprint arXiv:2405.17766 (2024).

In addition, it is a common practice when releasing a database (PTB-XL, CSPC2018 or Physionet Challenge 2017), to propose a split in order to avoid this issue.

**Questions:**

-

---

### Official Review · Reviewer_7i3o · 2024-11-08

**Soundness:** 2
**Presentation:** 2
**Contribution:** 1
**Rating:** 3
**Confidence:** 4

**Summary:**

This paper addresses the challenges posed by subject-specific features in MedTS evaluation, highlighting how inappropriate evaluation setups can lead to misleadingly high performance. The authors categorize evaluation setups into two types: subject-dependent and subject-independent, demonstrating that subject-independent setups are generally more suitable for various MedTS tasks.

In the conclusion, the authors introduce a new taxonomy for MedTS data, breaking it down into disease-related features, subject-specific features, and extraneous features. They propose three experimental setups to examine the influence of subject-specific features on model performance. Their experiments show that models can achieve artificially high accuracy in subject-dependent setups without learning disease-related features, supporting the recommendation that subject-independent evaluation is a more reliable approach for MedTS tasks.

**Strengths:**

1. Emphasizing the Importance of Evaluation Protocols

Developing a novel algorithm is certainly of interest to researchers in this field, but focusing on creating better and more useful evaluation protocols may be even more important for the algorithm’s effectiveness. This is especially crucial in the medical field. In this regard, this work could be beneficial to the research community. Additionally, I find it interesting that, after they completely removed disease-related features across subjects in MedFormer, the performance did not drop significantly. This suggests that the model may be learning spurious correlations in the dataset, which are not related to the true underlying values.

2. Experimental Design for Hypothesis Validation

Section 3.3 introduces three experiments—subject discrimination, random-label subject-dependent, and random-label subject-independent—to demonstrate that models can exploit spurious correlations to improve performance in subject-dependent settings. This approach is reasonable and likely to support the hypothesis.

**Weaknesses:**

1. Limited Novelty on the Audience in Medical Fields

The primary motivation behind this work is to encourage researchers to focus on selecting a more appropriate evaluation protocol, which I commend. However, the proposed "subject-independent evaluation protocol" is already a common practice in the evaluation of many machine learning algorithms within clinical journals. The examples listed below are not cherry-picked through extensive literature searches; with minimal effort, one can find numerous papers where evaluations are conducted in a subject-independent manner.
- Figure 1 in “Prospective, Multi-site Study of Patient Outcomes After Implementation of the TREWS Machine Learning-based Early Warning System for Sepsis,” R. Adams et al., Nature Medicine, 2022.
- Figure 1 in “Development and Validation of a Deep Learning Model to Reduce the Interference of Rectal Artifacts in MRI-based Prostate Cancer Diagnosis,” L. Hu et al., Radiology, 2024.
You may want to refer to the cited papers above to check whether this subject-independent evaluation protocol is indeed widely accepted.

2. Absence of Type-II Experiments

In real-world scenarios, Type-II (multi-subject and multi-class) problems are very common. Therefore, to enhance the quality of this research project, it would be beneficial to include extensive experiments on Type-II datasets.

**Questions:**

1. “Researchers should use the subject-independent setup when evaluating MedTS datasets”.

The word "should" may be too strong here. When developing an adaptive algorithm for a specific patient, the subject-dependent evaluation protocol may also be appropriate.

2. “However, many studies are still evaluating MedTS in subject-dependent setup, which is problematic.”

This assertion might not be entirely accurate for the medical field, as many studies already use subject-independent setups. Please refer to my references in the weaknesses section for examples.

---

### Note · Authors · 2024-11-13

**Comment:**

We sincerely thank all the reviewers for their time and effort in evaluating our paper. We find many of the reviews to be fair and constructive, and we appreciate the valuable feedback. Due to the limited time available for the rebuttal, we have decided to withdraw the paper and focus on making comprehensive revisions for a stronger paper version in the future. Thanks again!

**Withdrawal Confirmation:**

I have read and agree with the venue's withdrawal policy on behalf of myself and my co-authors.